# An Autoencoder Approach to Learning Bilingual Word Representations

**Sarath Chandar A P**[1] *, **Stanislas Lauly**[2] *, **Hugo Larochelle**[2], **Mitesh M Khapra**[3],
**Balaraman Ravindran**[1], **Vikas Raykar**[3], **Amrita Saha**[3]
[1]Indian Institute of Technology Madras, [2]Université de Sherbrooke, [3]IBM Research India
apsarathchandar@gmail.com, {stanislas.lauly,hugo.larochelle}@usherbrooke.ca,
{mikhapra,viraykar,amrsaha4}@in.ibm.com, ravi@cse.iitm.ac.in

## Abstract

Cross-language learning allows one to use training data from one language to build models for a different language. Many approaches to bilingual learning require that we have word-level alignment of sentences from parallel corpora. In this work we explore the use of autoencoder-based methods for cross-language learning of vectorial word representations that are coherent between two languages, while not relying on word-level alignments. We show that by simply learning to reconstruct the bag-of-words representations of aligned sentences, within and between languages, we can in fact learn high-quality representations and do without word alignments. We empirically investigate the success of our approach on the problem of cross-language text classification, where a classifier trained on a given language (*e.g.*, English) must learn to generalize to a different language (*e.g.*, German). In experiments on 3 language pairs, we show that our approach achieves state-of-the-art performance, outperforming a method exploiting word alignments and a strong machine translation baseline.

## 1 Introduction

The accuracy of Natural Language Processing (NLP) tools for a given language depend heavily on the availability of annotated resources in that language. For example, high quality POS taggers [1], parsers [2], sentiment analyzers [3] are readily available for English. However, this is not the case for many other languages such as Hindi, Marathi, Bodo, Farsi, and Urdu, for which annotated data is scarce. This situation was acceptable in the past when only a few languages dominated the digital content available online and elsewhere. However, the ever increasing number of languages on the web today has made it important to accurately process natural language data in such resource-deprived languages also. An obvious solution to this problem is to improve the annotated inventory of these languages, but the cost, time and effort required act as a natural deterrent to this.

Another option is to exploit the unlabeled data available in a language. In this context, vectorial text representations have proven useful for multiple NLP tasks [4, 5]. It has been shown that meaningful representations, capturing syntactic and semantic similarity, can be learned from unlabeled data. While the majority of previous work on vectorial text representations has concentrated on the monolingual case, there has also been considerable interest in learning word and document representations that are aligned across languages [6, 7, 8, 9, 10, 11, 12]. Such aligned representations allow the use of resources from a resource-fortunate language to develop NLP capabilities in a resource-deprived language.

One approach to cross-lingual exploitation of resources is to project parameters learned from the annotated data of one language to another language [13, 14, 15, 16, 17]. These approaches rely on a

bilingual resource such as a Machine Translation (MT) system. Recent attempts at learning common bilingual representations [9, 10, 11] aim to eliminate the need of such an MT system. A common property of these approaches is that a *word-level alignment* of translated sentences is leveraged to derive a regularization term relating word embeddings across languages. Such methods not only eliminate the need for an MT system but also outperform MT based projection approaches.

In this paper, we experiment with methods that learn bilingual word representations *without word-to-word alignments* of bilingual corpora during training. Unlike previous approaches, we only require aligned sentences and do not rely on word-level alignments (e.g., extracted using GIZA++, as is usual), simplifying the learning procedure. To do so, we propose and investigate bilingual autoencoder models, that learn hidden encoder representations of paired bag-of-words sentences that are not only informative of the original bag-of-words but also predictive of the other language. Word representations can then easily be extracted from the encoder and used in the context of a supervised NLP task. Specifically, we demonstrate the quality of these representations for the task of cross-language document classification, where a labeled data set can be available in one language, but not in another one. As we'll see, our approach is able to reach state-of-the-art performance, outperforming a method exploiting word alignments and a strong machine translation baseline.

## 2  Autoencoder for Bags-of-Words

Let $\mathbf{x}$ be the bag-of-words representation of a sentence. Specifically, each $x_i$ is a word index from a fixed vocabulary of $V$ words. As this is a bag-of-words, the order of the words within $\mathbf{x}$ does not correspond to the word order in the original sentence. We wish to learn a $D$-dimensional vectorial representation of our words from a training set of sentence bags-of-words $\{\mathbf{x}^{(t)}\}_{t=1}^{T}$.

We propose to achieve this by using an autoencoder model that encodes an input bag-of-words $\mathbf{x}$ with a sum of the representations (embeddings) of the words present in $\mathbf{x}$, followed by a non-linearity. Specifically, let matrix $\mathbf{W}$ be the $D \times V$ matrix whose columns are the vector representations for each word. The encoder's computation will involve summing over the columns of $\mathbf{W}$ for each word in the bag-of-word. We will denote this encoder function $\phi(\mathbf{x})$. Then, using a decoder, the autoencoder will be trained to optimize a loss function that measures how predictive of the original bag-of-words is the encoder representation $\phi(\mathbf{x})$.

There are different variations we can consider in the design of the encoder/decoder and the choice of loss function. One must be careful however, as certain choices can be inappropriate for training on word observations, which are intrinsically sparse and high-dimensional. In this paper, we explore and compare two different approaches, described in the next two sub-sections.

### 2.1  Binary bag-of-words reconstruction training with merged bags-of-words

In the first approach, we start from the conventional autoencoder architecture, which minimizes a cross-entropy loss that compares a binary vector observation with a decoder reconstruction. We thus convert the bag-of-words $\mathbf{x}$ into a fixed-size but sparse binary vector $\mathbf{v}(\mathbf{x})$, which is such that $v(\mathbf{x})_{x_i}$ is 1 if word $x_i$ is present in $\mathbf{x}$ and otherwise 0.

From this representation, we obtain an encoder representation by multiplying $\mathbf{v}(\mathbf{x})$ with the word representation matrix $\mathbf{W}$

$$\mathbf{a}(\mathbf{x}) = \mathbf{c} + \mathbf{W}\mathbf{v}(\mathbf{x}), \ \ \phi(\mathbf{x}) = \mathbf{h}(\mathbf{a}(\mathbf{x})) \tag{1}$$

where $\mathbf{h}(\cdot)$ is an element-wise non-linearity such as the sigmoid or hyperbolic tangent, and $\mathbf{c}$ is a $D$-dimensional bias vector. Encoding thus involves summing the word representations of the words present at least once in the bag-of-words.

To produce a reconstruction, we parametrize the decoder using the following non-linear form:

$$\widehat{\mathbf{v}}(\mathbf{x}) = \mathrm{sigm}(\mathbf{V}\phi(\mathbf{x}) + \mathbf{b}) \tag{2}$$

where $\mathbf{V} = \mathbf{W}^T$, $\mathbf{b}$ is the bias vector of the reconstruction layer and $\mathrm{sigm}(a) = 1/(1 + \exp(-a))$ is the sigmoid non-linearity.

Then, the reconstruction is compared to the original binary bag-of-words as follows:

$$\ell(\mathbf{v}(\mathbf{x})) = -\sum_{i=1}^{V} v(\mathbf{x})_i \log(\widehat{v}(\mathbf{x})_i) + (1 - v(\mathbf{x})_i) \log(1 - \widehat{v}(\mathbf{x})_i) \ . \tag{3}$$

Training proceeds by optimizing the sum of reconstruction cross-entropies across the training set, *e.g.*, using stochastic or mini-batch gradient descent.

Note that, since the binary bags-of-words are very high-dimensional (the dimensionality corresponds to the size of the vocabulary, which is typically large), the above training procedure which aims at reconstructing the complete binary bag-of-word, will be slow. Since we will later be training on millions of sentences, training on each individual sentence bag-of-words will be expensive.

Thus, we propose a simple trick, which exploits the bag-of-words structure of the input. Assuming we are performing mini-batch training (where a mini-batch contains a list of the bags-of-words of adjacent sentences), we simply propose to merge the bags-of-words of the mini-batch into a single bag-of-words and perform an update based on that merged bag-of-words. The resulting effect is that each update is as efficient as in stochastic gradient descent, but the number of updates per training epoch is divided by the mini-batch size . As we'll see in the experimental section, this trick produces good word representations, while sufficiently reducing training time. We note that, additionally, we could have used the stochastic approach proposed by Dauphin et al. [18] for reconstructing binary bag-of-words representations of documents, to further improve the efficiency of training. They use importance sampling to avoid reconstructing the whole $V$-dimensional input vector.

## 2.2 Tree-based decoder training

The previous autoencoder architecture worked with a binary vectorial representation of the input bag-of-words. In the second autoencoder architecture we investigate, we consider an architecture that instead works with the bag (unordered list) representation more directly.

First, the encoder representation will now involve a sum of the representation of all words, reflecting the relative frequency of each word:

$$\mathbf{a}(\mathbf{x}) = \mathbf{c} + \sum_{i=1}^{|\mathbf{x}|} \mathbf{W}_{\cdot,x_i}, \ \ \boldsymbol{\phi}(\mathbf{x}) = \mathbf{h}\left(\mathbf{a}(\mathbf{x})\right) \ . \tag{4}$$

Moreover, decoder training will assume that, from the decoder's output, we can obtain a probability distribution $p(\widehat{x}|\boldsymbol{\phi}(\mathbf{x}))$ over any word $\widehat{x}$ observed at the reconstruction output layer. Then, we can treat the input bag-of-words as a $|\mathbf{x}|$-trials multinomial sample from that distribution and use as the reconstruction loss its negative log-likelihood:

$$\ell(\mathbf{x}) = \sum_{i=1}^{V} -\log p(\widehat{x} = x_i|\boldsymbol{\phi}(\mathbf{x})) \ . \tag{5}$$

We now must ensure that the decoder can compute $p(\widehat{x} = x_i|\boldsymbol{\phi}(\mathbf{x}))$ efficiently from $\boldsymbol{\phi}(\mathbf{x})$. Specifically, we'd like to avoid a procedure scaling linearly with the vocabulary size $V$, since $V$ will be very large in practice. This precludes any procedure that would compute the numerator of $p(\widehat{x} = w|\boldsymbol{\phi}(\mathbf{x}))$ for each possible word $w$ separately and normalize it so it sums to one.

We instead opt for an approach borrowed from the work on neural network language models [19, 20]. Specifically, we use a probabilistic tree decomposition of $p(\widehat{x} = x_i|\boldsymbol{\phi}(\mathbf{x}))$. Let's assume each word has been placed at the leaf of a binary tree. We can then treat the sampling of a word as a stochastic path from the root of the tree to one of the leaves.

We denote as $\mathbf{l}(x)$ the sequence of internal nodes in the path from the root to a given word $x$, with $l(x)_1$ always corresponding to the root. We will denote as $\boldsymbol{\pi}(x)$ the vector of associated left/right branching choices on that path, where $\pi(x)_k = 0$ means the path branches left at internal node $l(x)_k$ and otherwise branches right if $\pi(x)_k = 1$. Then, the probability $p(\widehat{x} = x|\boldsymbol{\phi}(\mathbf{x}))$ of reconstructing a certain word $x$ observed in the bag-of-words is computed as

$$p(\widehat{x}|\boldsymbol{\phi}(\mathbf{x})) = \prod_{k=1}^{|\boldsymbol{\pi}(\hat{x})|} p(\pi(\widehat{x})_k|\boldsymbol{\phi}(\mathbf{x})) \tag{6}$$

where $p(\pi(\hat{x})_k | \phi(\mathbf{x}))$ is output by the decoder. By using a full binary tree of words, the number of different decoder outputs required to compute $p(\hat{x} | \phi(\mathbf{x}))$ will be logarithmic in the vocabulary size $V$. Since there are $|\mathbf{x}|$ words in the bag-of-words, at most $O(|\mathbf{x}| \log V)$ outputs are required from the decoder. This is of course a worst case scenario, since words will share internal nodes between their paths, for which the decoder output can be computed just once. As for organizing words into a tree, as in Larochelle and Lauly [21] we used a random assignment of words to the leaves of the full binary tree, which we have found to work well in practice.

Finally, we need to choose a parametrized form for the decoder. We choose the following form:

$$p(\pi(\hat{x})_k = 1 | \phi(\mathbf{x})) = \mathrm{sigm}(b_{l(\hat{x}_i)_k} + \mathbf{V}_{l(\hat{x}_i)_k,.} \phi(\mathbf{x})) \tag{7}$$

where $\mathbf{b}$ is a $(V\text{-}1)$-dimensional bias vector and $\mathbf{V}$ is a $(V-1) \times D$ matrix. Each left/right branching probability is thus modeled with a logistic regression model applied on the encoder representation of the input bag-of-words $\phi(\mathbf{x})$.

## 3 Bilingual autoencoders

Let's now assume that for each sentence bag-of-words $\mathbf{x}$ in some source language $\mathcal{X}$, we have an associated bag-of-words $\mathbf{y}$ for this sentence translated in some target language $\mathcal{Y}$ by a human expert.

Assuming we have a training set of such $(\mathbf{x}, \mathbf{y})$ pairs, we'd like to use it to learn representations in both languages that are aligned, such that pairs of translated words have similar representations.

To achieve this, we propose to augment the regular autoencoder proposed in Section 2 so that, from the sentence representation in a given language, a reconstruction can be attempted of the original sentence in the other language. Specifically, we now define language specific word representation matrices $\mathbf{W}^x$ and $\mathbf{W}^y$, corresponding to the languages of the words in $\mathbf{x}$ and $\mathbf{y}$ respectively. Let $V^{\mathcal{X}}$ and $V^{\mathcal{Y}}$ also be the number of words in the vocabulary of both languages, which can be different. The word representations however are of the same size $D$ in both languages. For the binary reconstruction autoencoder, the bag-of-words representations extracted by the encoder become

$$\phi(\mathbf{x}) = \mathbf{h}\left(\mathbf{c} + \mathbf{W}^{\mathcal{X}}\mathbf{v}(\mathbf{x})\right) \ , \ \ \phi(\mathbf{y}) = \mathbf{h}\left(\mathbf{c} + \mathbf{W}^{\mathcal{Y}}\mathbf{v}(\mathbf{y})\right)$$

and are similarly extended for the tree-based autoencoder. Notice that we share the bias $\mathbf{c}$ before the non-linearity across encoders, to encourage the encoders in both languages to produce representations on the same scale.

From the sentence in either languages, we want to be able to perform a reconstruction of the original sentence in both the languages. In particular, given a representation in any language, we'd like a decoder that can perform a reconstruction in language $\mathcal{X}$ and another decoder that can reconstruct in language $\mathcal{Y}$. Again, we use decoders of the form proposed in either Section 2.1 or 2.2 (see Figure 1), but let the decoders of each language have their own parameters $(\mathbf{b}^{\mathcal{X}}, \mathbf{V}^{\mathcal{X}})$ and $(\mathbf{b}^{\mathcal{Y}}, \mathbf{V}^{\mathcal{Y}})$.

This encoder/decoder decomposition structure allows us to learn a mapping within each language and across the languages. Specifically, for a given pair $(\mathbf{x}, \mathbf{y})$, we can train the model to (1) construct $\mathbf{y}$ from $\mathbf{x}$ (loss $\ell(\mathbf{x}, \mathbf{y})$), (2) construct $\mathbf{x}$ from $\mathbf{y}$ (loss $\ell(\mathbf{y}, \mathbf{x})$), (3) reconstruct $\mathbf{x}$ from itself (loss $\ell(\mathbf{x})$) and (4) reconstruct $\mathbf{y}$ from itself (loss $\ell(\mathbf{y})$). We follow this approach in our experiments and optimize the sum of the corresponding 4 losses during training.

### 3.1 Joint reconstruction and cross-lingual correlation

We also considered incorporating two additional terms to the loss function, in an attempt to favour even more meaningful bilingual representations:

$$\ell(\mathbf{x}, \mathbf{y}) + \ell(\mathbf{y}, \mathbf{x}) + \ell(\mathbf{x}) + \ell(\mathbf{y}) + \beta\ell([\mathbf{x}, \mathbf{y}], [\mathbf{x}, \mathbf{y}]) - \lambda \cdot cor(\mathbf{a}(\mathbf{x}), \mathbf{a}(\mathbf{y})) \tag{8}$$

The term $\ell([\mathbf{x}, \mathbf{y}], [\mathbf{x}, \mathbf{y}])$ is simply a joint reconstruction term, where both languages are simultanouesly presented as input and reconstructed. The second term $cor(\mathbf{a}(\mathbf{x}), \mathbf{a}(\mathbf{y}))$ encourages correlation between the representation of each language. It is the sum of the scalar correlations between each pair $a(\mathbf{x})_k, a(\mathbf{y})_k$, across all dimensions $k$ of the vectors $\mathbf{a}(\mathbf{x}), \mathbf{a}(\mathbf{y})$[1]. To obtain a stochastic estimate of the correlation, during training, small mini-batches are used.

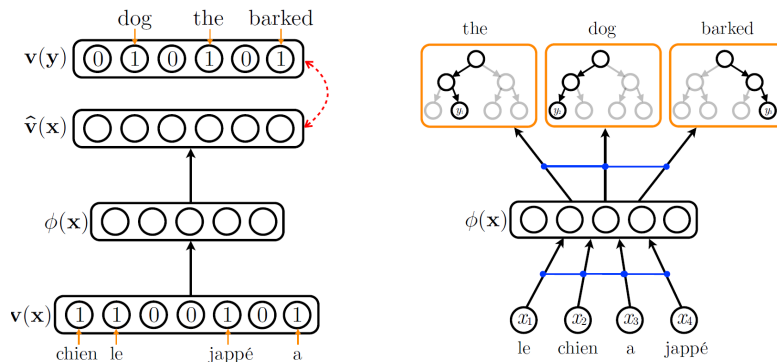

Figure 1: **Left:** Bilingual autoencoder based on the binary reconstruction error. **Right:** Tree-based bilingual autoencoder. In this example, they both reconstruct the bag-of-words for the English sentence "*the dog barked*" from its French translation "*le chien a jappé*".

## 3.2 Document representations

Once we learn the language specific word representation matrices $\mathbf{W}^x$ and $\mathbf{W}^y$ as described above, we can use them to construct document representations, by using their columns as word vector representations. Given a document $\mathbf{d}$ written in language $\mathcal{Z} \in \{\mathcal{X}, \mathcal{Y}\}$ and containing $m$ words, $z_1, z_2, \ldots, z_m$, we represent it as the tf-idf weighted sum of its words' representations $\psi(\mathbf{d}) = \sum_{i=1}^{m} \textit{tf-idf}(z_i) \cdot \mathbf{W}^{\mathcal{Z}}_{\cdot, z_i}$. We use the document representations thus obtained to train our document classifiers, in the cross-lingual document classification task described in Section 5.

## 4 Related Work

Recent work that has considered the problem of learning bilingual representations of words usually has relied on word-level alignments. Klementiev et al. [9] propose to train simultaneously two neural network languages models, along with a regularization term that encourages pairs of frequently aligned words to have similar word embeddings. Thus, the use of this regularization term requires to first obtain word-level alignments from parallel corpora. Zou et al. [10] use a similar approach, with a different form for the regularizer and neural network language models as in [5]. In our work, we specifically investigate whether a method that does not rely on word-level alignments can learn comparably useful multilingual embeddings in the context of document classification.

Looking more generally at neural networks that learn multilingual representations of words or phrases, we mention the work of Gao et al. [22] which showed that a useful linear mapping between *separately trained* monolingual skip-gram language models could be learned. They too however rely on the specification of pairs of words in the two languages to align. Mikolov et al. [11] also propose a method for training a neural network to learn useful representations of phrases, in the context of a phrase-based translation model. In this case, phrase-level alignments (usually extracted from word-level alignments) are required. Recently, Hermann and Blunsom [23], [24] proposed neural network architectures and a margin-based training objective that, as in this work, does not rely on word alignments. We will briefly discuss this work in the experiments section.

## 5 Experiments

The techniques proposed in this paper enable us to learn bilingual embeddings which capture cross-language similarity between words. We propose to evaluate the quality of these embeddings by using them for the task of cross-language document classification. We followed closely the setup used by Klementiev et al. [9] and compare with their method, for which word representations are publicly available[2]. The set up is as follows. A labeled data set of documents in some language $\mathcal{X}$ is available to train a classifier, however we are interested in classifying documents in a different language $\mathcal{Y}$ at test time. To achieve this, we leverage some bilingual corpora, which is not labeled with any

document-level categories. This bilingual corpora is used to learn document representations that are coherent between languages $\mathcal{X}$ and $\mathcal{Y}$. The hope is thus that we can successfully apply the classifier trained on document representations for language $\mathcal{X}$ directly to the document representations for language $\mathcal{Y}$. Following this setup, we performed experiments on 3 data sets of language pairs: English/German (EN/DE), English/French (EN/FR) and English/Spanish (EN/ES).

## 5.1 Data

For learning the bilingual embeddings, we used sections of the Europarl corpus [25] which contains roughly 2 million parallel sentences. We considered 3 language pairs. We used the same pre-processing as used by Klementiev et al. [9]. We tokenized the sentences using NLTK [26], removed punctuations and lowercased all words. We did not remove stopwords.

As for the labeled document classification data sets, they were extracted from sections of the Reuters RCV1/RCV2 corpora, again for the 3 pairs considered in our experiments. Following Klementiev et al. [9], we consider only documents which were assigned exactly one of the 4 top level categories in the topic hierarchy (CCAT, ECAT, GCAT and MCAT). These documents are also pre-processed using a similar procedure as that used for the Europarl corpus. We used the same vocabularies as those used by Klementiev et al. [9] (varying in size between $35,000$ and $50,000$).

For each pair of languages, our overall procedure for cross-language classification can be summarized as follows:

**Train representation:** Train bilingual word representations $\mathbf{W}^x$ and $\mathbf{W}^y$ on sentence pairs extracted from Europarl for languages $\mathcal{X}$ and $\mathcal{Y}$. Optionally, we also use the monolingual documents from RCV1/RCV2 to reinforce the monolingual embeddings (this choice is cross-validated). These non-parallel documents can be used through the losses $\ell(\mathbf{x})$ and $\ell(\mathbf{y})$ (*i.e.* by reconstructing $\mathbf{x}$ from $\mathbf{x}$ or $\mathbf{y}$ from $\mathbf{y}$). Note that Klementiev et al. [9] also used this data when training word representations.

**Train classifier:** Train document classifier on the Reuters training set for language $\mathcal{X}$, where documents are represented using the word representations $\mathbf{W}^x$ (see Section 3.2). As in Klementiev et al. [9] we used an averaged perceptron trained for 10 epochs, for all the experiments.

**Test-time classification:** Use the classifier trained in the previous step on the Reuters test set for language $\mathcal{Y}$, using the word representations $\mathbf{W}^y$ to represent the documents.

We trained the following autoencoders[3]: **BAE-cr** which uses reconstruction error based decoder training (see Section 2.1) and **BAE-tr** which uses tree-based decoder training (see Section 2.2).

Models were trained for up to 20 epochs using the same data as described earlier. BAE-cr used mini-batch (of size 20) stochastic gradient descent, while BAE-tr used regular stochastic gradient. All results are for word embeddings of size $D = 40$, as in Klementiev et al. [9]. Further, to speed up the training for BAE-cr we merged each 5 adjacent sentence pairs into a single training instance, as described in Section 2.1. For all language pairs, the joint reconstruction $\beta$ was fixed to $1$ and the cross-lingual correlation factor $\lambda$ to $4$ for BAE-cr. For BAE-tr, none of these additional terms were found to be particularly beneficial, so we set their weights to 0 for all tasks. The other hyper-parameters were tuned to each task using a training/validation set split of 80% and 20% and using the performance on the validation set of an averaged perceptron trained on the smaller training set portion (notice that this corresponds to a monolingual classification experiment, since the general assumption is that no labeled data is available in the test set language).

## 5.2 Comparison of the performance of different models

We now present the cross language classification results obtained by using the embeddings produced by our two autoencoders. We also compare our models with the following approaches:

**Klementiev et al.**: This model uses word embeddings learned by a multitask neural network language model with a regularization term that encourages pairs of frequently aligned words to have similar word embeddings. From these embeddings, document representations are computed as described in Section 3.2.

Table 1: Cross-lingual classification accuracy for 3 language pairs, with 1000 labeled examples.

|  | EN → DE | DE → EN | EN → FR | FR → EN | EN → ES | ES → EN |
|---|---|---|---|---|---|---|
| BAE-tr | 81.8 | 60.1 | 70.4 | 61.8 | **59.4** | 60.4 |
| BAE-cr | **91.8** | **74.2** | **84.6** | **74.2** | 49.0 | **64.4** |
| Klementiev et al. | 77.6 | 71.1 | 74.5 | 61.9 | 31.3 | 63.0 |
| MT | 68.1 | 67.4 | 76.3 | 71.1 | 52.0 | 58.4 |
| Majority Class | 46.8 | 46.8 | 22.5 | 25.0 | 15.3 | 22.2 |

Table 2: Example English words along with the closest words both in English (EN) and German (DE), using the Euclidean distance between the embeddings learned by BAE-cr.

| Word | Lang | Nearest neighbors | Word | Lang | Nearest neighbors |
|---|---|---|---|---|---|
| january | EN | january, march, october | oil | EN | oil, supply, supplies, gas |
|  | DE | januar, märz, oktober |  | DE | öl, boden, befindet, gerät |
| president | EN | president, i, mr, presidents | microsoft | EN | microsoft, cds, insider |
|  | DE | präsident, präsidentin |  | DE | microsoft, cds, warner |
| said | EN | said, told, say, believe | market | EN | market, markets, single |
|  | DE | gesagt, sagte, sehr, heute |  | DE | markt, marktes, märkte |

**MT:** Here, test documents are translated to the language of the training documents using a standard phrase-based MT system, MOSES[4] which was trained using default parameters and a 5-gram language model on the Europarl corpus (same as the one used for inducing our bilingual embeddings).

**Majority Class:** Test documents are simply assigned the most frequent class in the training set.

For the EN/DE language pairs, we directly report the results from Klementiev et al. [9]. For the other pairs (not reported in Klementiev et al. [9]), we used the embeddings available online and performed the classification experiment ourselves. Similarly, we generated the MT baseline ourselves.

Table 1 summarizes the results. They were obtained using 1000 RCV training examples. We report results in both directions, *i.e.* language $\mathcal{X}$ to $\mathcal{Y}$ and vice versa. The best performing method is always either BAE-cr or BAE-tr, with BAE-cr having the best performance overall. In particular, BAE-cr often outperforms the approach of Klementiev et al. [9] by a large margin.

We also mention the recent work of Hermann and Blunsom [23], who proposed two neural network architectures for learning word and document representations using sentence-aligned data only. Instead of an autoencoder paradigm, they propose a margin-based objective that aims to make the representation of aligned sentences closer than non-aligned sentences. While their trained embeddings are not publicly available, they report results for the EN/DE classification experiments, with representations of the same size as here ($D = 40$) and trained on 500K EN/DE sentence pairs. Their best model reaches accuracies of 83.7% and 71.4% respectively for the EN → DE and DE → EN tasks. One clear advantage of our model is that unlike their model, it can use additional monolingual data. Indeed, when we train BAE-cr with 500k EN/DE sentence pairs, plus monolingual RCV documents (which come at no additional cost), we get accuracies of 87.9% (EN → DE) and 76.7% (DE → EN), still improving on their best model. If we do not use the monolingual data, BAE-cr's performance is worse but still competitive at 86.1% for EN → DE and 68.8% for DE → EN.

We also evaluate the effect of varying the amount of supervised training data for training the classifier. For brevity, we report only the results for the EN/DE pair, which are summarized in Figure 2. We observe that BAE-cr clearly outperforms the other models at almost all data sizes. More importantly, it performs remarkably well at very low data sizes (100), suggesting it learns very meaningful embeddings, though the method can still benefit from more labeled data (as in the DE → EN case).

Table 2 also illustrates the properties captured within and across languages, for the EN/DE pair[5]. For a few English words, the words with closest word representations (in Euclidean distance) are shown, for both English and German. We observe that words that form a translation pair are close, but also that close words within a language are syntactically/semantically similar as well.

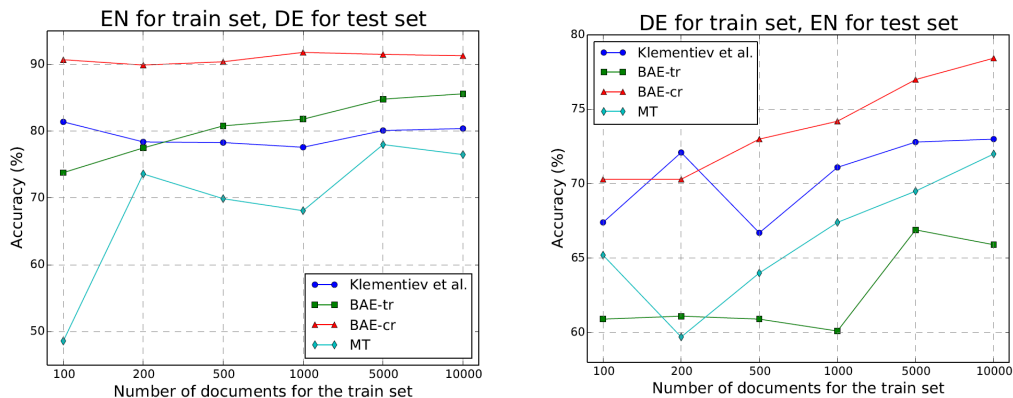

Figure 2: Cross-lingual classification accuracy results, from EN $\rightarrow$ DE (**left**), and DE $\rightarrow$ EN (**right**).

The excellent performance of BAE-cr suggests that merging several sentences into single bags-of-words can still yield good word embeddings. In other words, not only we do not need to rely on word-level alignments, but exact sentence-level alignment is also not essential to reach good performances. We experimented with the merging of 5, 25 and 50 adjacent sentences (see the supplementary material). Generally speaking, these experiments also confirm that even coarser merges can sometimes not be detrimental. However, for certain language pairs, there can be an important decrease in performance. On the other hand, when comparing the performance of BAE-tr with the use of 5-sentences merges, no substantial impact is observed.

# 6 Conclusion and Future Work

We presented evidence that meaningful bilingual word representations could be learned without relying on word-level alignments or using fairly coarse sentence-level alignments. In particular, we showed that even though our model does not use word level alignments, it is able to reach state-of-the-art performance, even compared to a method that exploits word-level alignments. In addition, it also outperforms a strong machine translation baseline.

For future work, we would like to investigate extensions of our bag-of-words bilingual autoencoder to bags-of-n-grams, where the model would also have to learn representations for short phrases. Such a model should be particularly useful in the context of a machine translation system. We would also like to explore the possibility of converting our bilingual model to a multilingual model which can learn common representations for multiple languages given different amounts of parallel data between these languages.

## Acknowledgement

We would like to thank Alexander Klementiev and Ivan Titov for providing the code for the classifier and data indices. This work was supported in part by Google.

## Footnotes

[1] While we could have applied the correlation term on $\phi(\mathbf{x}), \phi(\mathbf{y})$ directly, applying it to the pre-activation function vectors was found to be more numerically stable.

[2]http://people.mmci.uni-saarland.de/~aklement/data/distrib/

[3]Our word representations and code are available at `http://www.sarathchandar.in/crl.html`

[4]http://www.statmt.org/moses/

[5]See also the supplementary material for a t-SNE visualization of the word representations.

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
