[Supplementary Material]

# Supplementary Material: An Autoencoder Approach to Learning Bilingual Word Representations

**Sarath Chandar A P**[1] **\*, Stanislas Lauly**[2] **\*, Hugo Larochelle**[2]**, Mitesh M Khapra**[3]**,**
**Balaraman Ravindran**[1]**, Vikas Raykar**[3]**, Amrita Saha**[3]
[1]Indian Institute of Technology Madras, [2]Université de Sherbrooke, [3]IBM Research India
apsarathchandar@gmail.com, {stanislas.lauly,hugo.larochelle}@usherbrooke.ca,
{mikhapra,viraykar,amrsaha4}@in.ibm.com, ravi@cse.iitm.ac.in

## Abstract

We provide here additional details relatively to our paper.

## 1 Coarser alignments

We experimented with the merging of 5, 25 and 50 adjacent sentences into a single bag-of-words. Results are shown in Table 1. They suggest that merging several sentences into single bags-of-words does not necessarily impact the quality of the word embeddings. Thus they confirm that exact sentence-level alignment is not essential to reach good performances as well.

Table 1: Cross-lingual classification accuracy for 3 different pairs of languages, when merging the bag-of-words for different numbers of sentences. These results are based on 1000 labeled examples.

|         | # sent. | EN $\rightarrow$ DE | DE $\rightarrow$ EN | EN $\rightarrow$ FR | FR $\rightarrow$ EN | EN $\rightarrow$ ES | ES $\rightarrow$ EN |
|---------|---------|---------|---------|---------|---------|---------|---------|
| BAE-tr  | 1       | 81.8    | 60.1    | 70.4    | 61.8    | 59.4    | 60.4    |
|         | 5       | 84.0    | 67.7    | 72.08   | 65.7    | 58.325  | 54.48   |
|         | 25      | 83.0    | 63.4    | 73.92   | 59.48   | 41.7    | 52.2    |
|         | 50      | 75.9    | 68.6    | 73.96   | 62.34   | 46.35   | 47.22   |
| BAE-cr  | 5       | 91.75   | 72.78   | 84.64   | 74.2    | 49.02   | 64.4    |
|         | 25      | 88.0    | 64.5    | 78.1    | 70.02   | 68.3    | 54.68   |
|         | 50      | 90.2    | 49.2    | 82.44   | 75.5    | 38.2    | 67.38   |

## 2 Visualization of the word representations

In Figures 1 and 2, we present a 2D visualization of the word embeddings for the language pair English/German, generated using the t-SNE dimensionality reduction algorithm [1], for the BAE-cr and BAE-tr models. We see that words with similar meanings are close to each other, for words in different languages *and* for words within the same language. This confirms that these models were able to learn a meaningful semantic representation for the words.

## References

[1] Laurens van der Maaten and Geoffrey E Hinton. Visualizing Data using t-SNE. *Journal of Machine Learning Research*, 9:2579–2605, 2008. URL http://www.jmlr.org/papers/volume9/vandermaaten08a/vandermaaten08a.pdf.

Figure 1: For the BAE-cr model, a t-SNE 2D visualization of the learned English/German word representations (better visualized on a computer). Words hyphenated with "EN" and "DE" are English and German words respectively.

Figure 2: For the BAE-tr model, a t-SNE 2D visualization of the learned English/German word representations (better visualized on a computer). Words hyphenated with "EN" and "DE" are English and German words respectively.