[Reviews · NeurIPS 2014]

Submitted by Assigned_Reviewer_27

This paper proposes to learn bilingual word vector representations through an autoencoder.
The novelty of this approach is to not rely on word-level alignments. It only requires aligned sentences.
An autoencoder model is used to reconstruct the bag-of-words representation of aligned sentences, within and between languages.
Two different variations are proposed. The first one is based on binary bag-of-words (i.e. words present or not) and the second considers the frequency of each word.
The authors demonstrate the quality of these representations for the task of cross-language document classification, where they have been able to reach state-of-the-art performance.

I like the idea to learn cross-lingual word representations while not relying on word-level alignment.
This offers a clear advantage compared to other approaches.
Furthermore, the method using binary bag-of-words reconstruction is quite simple and give the best overall results. And I like simple methods.
However, the authors introduce a training with merged bags-of-words that it is not very clear to me.
They said that they merged adjacent sentences to perform mini-batch training and speed up the process.
But they don't say how they define sentences as adjacent.
This model is also vocabulary-dependent. The vocabulary size in the experiment is relatively small (between 35,000 and 50,000). Would this method be scalable with a larger vocabulary?
I am also surprised to see that the tree-based model performs significantly worse (less than 10-14%) for EN/DE, DE/EN, EN/FR and FR/EN pairs, while it performs much better for EN/ES (more than 10%).
How do the authors explain such variation?
That makes me think that the tree-based model is not properly trained.

In my opinion, it remains too many open questions in the experiment section.
The authors compared their results to Hermann and Blunsom's paper.
Hermann and Blusom also proposed a method to learn multilingual distributed representations without word alignment.
They showed in their paper the usefulness of adding training data to increase the overall performance.
However the training size is never mentioned in this paper. The authors only said that the corpus contains roughly 2 million parallel sentences. Did they train the word representations with all of them? In this case, it would make the training data twice larger than the Hermann and Blunsom's training dataset and explain why they achieve better performance.
Or it could be due to the fact that the authors fine-tuned the monolingual embeddings on RCV1/RCV2.
To make a fair comparison, it would be nice to run experiments with the same number of training sentence pairs and without fine-tuning.
The authors also said that they consider only documents which were assigned exactly one of the 4 top level categories in the topic hierarchy.
This makes a 4-class classification problem. How is it possible to have a majority class accucary smaller than 25% for EN/FR, EN/ES and ES/EN pairs?
Results for the other models are also not described in enough detail.
For Klementiev's model with EN/DE language pairs, the authors report the results from the paper. But is it the same test set?
What classifier is used with the MT's model?
My intuition would be to use the translated words to construct a document representation with the model described in section 3.2.
But I am not sure this is the case here, because the authors again report results from Klementiev et al. for EN/DE and DE/EN pairs.
Authors also said that their embeddings are meaningful because they can generalize well even at low data sizes. It is however only true for EN/DE pairs, as we can see that the accuracy is still increasing with 10,000 training documents for DE/EN pairs.

Finally, I think that the authors should mention the paper "Multilingual Distributed Representations without Word Alignment" from Hermann and Blunsom in the related work section.
As their method looks very similar to the one proposed in this paper, it seems to me important to mention what are the differences and the benefits of the method compared to theirs.

-- After reading authors' comments: we still believe there are too many imprecisions in the experimental setup for a NIPS paper.
Summary: An interesting approach with promising results. However, the experimental setup is not clear enough. This leaves too many questions open.

Submitted by Assigned_Reviewer_28

This paper proposes methods for learning bilingual autoencoders, using a bilingual parallel corpus (sentence pairs) but does not need sentences to be word aligned. The best proposed model minimizes the loss of three terms -- the reconstruction errors between two languages in both directions and the correlation between the vectors of source and target languages. When evaluated on tasks of cross-lingual document classification (on 3 language pairs), the proposed approach shows better results compared to the previous work (including very recent one from [Hermann&Blunsom ACL-14]).

The paper is generally written clearly, and the proposed approach is very reasonable. Most designs are not particularly new but reuse of existing methods for the bilingual setting. One claimed advantage of the proposed method is that word alignment is not needed. However, I'm wondering whether this could be due to the fact that the evaluation task is document classification, where bag-of-words representation often results in reasonably good performance. The examples shown in Table 2 alleviates some of this concern, but ideally, testing this cross-lingual representation in a different NLP task (e.g., named entity recognition) may make it more convincing.

Another place that can be improved is the related work. Cross-lingual word embedding is not a new problem. The earliest approach can be at least dated to CLLSI [1], OPCA [2], CCA [3] and Siamese neural network approaches [4], as well as bilingual topic models all have been proposed. Although most of them consider cross-lingual document pairs (however, merging mini-matches of bag-of-words makes this work close to raw BoW document representation ), they should certainly be surveyed and ideally compared in this paper.

Minor:
Some references have updated versions published in ACL-2014, including [Gao et al.] and [Hermann & Blunsom]. It's better to cite the formal publications in the final version of this paper.

[1] Dumais et al. Automatic cross-language retrieval using latent semantic indexing. AAAI-1997 Spring Symposium.
[2] Platt et al. Translingual Document Representations from Discriminative Projections. EMNLP-2010.
[3] Faruqui & Dyer. Improving Vector Space Word Representations Using Multilingual Correlation. EACL-2014.
[4] Yih et al. Learning Discriminative Projections for Text Similarity Measures. CoNLL-2011.
Summary: + The proposed bilingual autoencoders are reasonably well designed and shows some improvement on a cross-lingual document classification task.
- Some experiments on tasks other than document classification may help validate the general effectiveness of the proposed word representations.
- The related work should include some discussion on the original work on cross-lingual embeddings.

Submitted by Assigned_Reviewer_37

This is a well written paper that proposes an autoencoder scheme for whole sentences (of bilingual text), which get encoded as a bag of words (represented as the sum of the word vectors of the words in the sentence).

I enjoyed reading the paper, and the methodology is simple and well explained. The authors do a fair job describing state of the art, and the problem they tackle is very important, as their method does not require multilingual aligned text. However, and unfortunately, a very related method recently appeared in ICLR (ref. 19). This, in my opinion, does not render the paper invalid - the fact that this idea came simultaneously is indicative of its relevance - but I'd like the authors to extend and discuss about their vs. ref. 19s approach - could they be cast equivalently? Why would you pick one or the other approach?
Summary: This is a good paper which had the misfortune of recent work that is very similar and was previously presented at ICLR. The authors approach, however, seems different enough, and the results support their method versus the recently published technique.

The only weakness of the paper is the lack of a more sophisticated method that is not "bag of words" like, which would enormously enrich the empirical results, and add another dimension of analysis currently lacking in this kind of literature.
Author Feedback
Author rebuttal: We thank the reviewers for their comments. We think we’re able here to address all of the main issues identified, especially those from Assigned_Reviewer_27. We’ll be happy to incorporate these precisions in our final draft, as they’ll certainly improve it.

The most important criticism is probably about comparing with Hermann and Blunsom. Assigned_Reviewer_27 asks: “To make a fair comparison, it would be nice to run experiments with the same number of training sentence pairs and without fine-tuning.”

The first request, i.e. to use the same number of sentences, is a good point. We performed this experiment and found that it does not change our main result. Indeed, using only 500k sentences from Europarl, we still achieve 87.9% for EN/DE and 76.7% for DE/EN, still improving on the Hermann and Blunsom performance of 83.7% and 71.4% respectively.

The second request, i.e. to not use the RCV Data, would actually be an unfair experiment, as it would correspond to asking us to “cripple” our model. Indeed, notice that the Hermann and Blunsom model *cannot* exploit unilingual data (they require aligned pairs of documents or sentences, as positive examples for their training objective). Since the RCV documents are not aligned, they cannot exploit them for learning word vectors. Our autoencoder however, has the advantage of being able to exploit such data in an unsupervised way, through the within-language reconstruction losses. Note also that such monolingual data comes at zero cost, compared to bilingual data. We actually really appreciate that Assigned_Reviewer_27 pointed this out, as this is another advantage of our model which we should have highlighted more. So we still performed this additional experiment (without the RCV data), to measure its impact. We obtained 86.1% for EN/DE and 68.8% for DE/EN. Thus, we are then still competitive with the Hermann and Blunsom model, beating it for EN/DE. This ability of our model to use unlabeled monolingual data is thus indeed useful, and we’ll be happy to report this insight in the final draft.

We wish to emphasize that it would be misleading to refer to our use of the RCV documents as fine-tuning, since fine-tuning normally refers to the use of *labeled* data. When we learn word vectors, we DO NOT use the label information from the RCV documents. Assigned_Reviewer_27 might have missed this, so we wanted to make sure this point was clear.

”But they don't say how they define sentences as adjacent.”
Europarl sentences come in the order they appear in the original texts. By adjacent, we mean sentences that appear next to each other in that order.

“Would this method be scalable with a larger vocabulary?”
The tree-based model is logarithmic in the vocabulary size, so it does scale. As for the binary autoencoder, we could use the reconstruction sampling method proposed by Dauphin et al. (2011) (Large-Scale Learning of Embeddings with Reconstruction Sampling), which exploits the sparsity in the data to scale up to large vocabularies.

Regarding all questions about the use of results from Klementiev et al., we actually contacted them before performing our experiments and obtained both the data splits and the averaged perceptron code they used. So the results are comparable. Our intention was to build upon their work, so we followed their experimental setup closely (data splits, baselines, etc.), as did Hermann and Blunsom.

“ How is it possible to have a majority class accucary smaller than 25%”
Note that in Cross Language classification, the majority class in one language (e.g. training) need not be the same as the majority class in another language (e.g. testing). This is what happened in these experiments.

“That makes me think that the tree-based model is not properly trained.”
We were very careful in training the tree-based model (e.g. trying different learning rates and following other practical guidelines as reported in the literature). The same practices were followed for training both models. Hopefully our SOTA results with the binary autoencoder will convince the reviewer that our neural networks were indeed trained properly.

“Authors also said that their embeddings are meaningful because they can generalize well even at low data sizes. It is however only true for EN/DE pairs, as we can see that the accuracy is still increasing with 10,000 training documents for DE/EN pairs.”
This is indeed misleading. We meant it still had good performance, even if a bit worse than with a larger dataset. We’ll fix this.

Finally, there were many relevant references mentioned by the reviewers. We’ll of course be happy to add them to the paper and discuss them in our related work section.